

# Genome-wide identification of long non-coding RNAs in tomato plants irradiated by neutrons followed by infection with *Tomato yellow leaf curl virus*

Yujie Zhou[1,*], Won Kyong Cho[2,*], Hee-Seong Byun[3], Vivek Chavan[1], Eui-Joon Kil[3], Sukchan Lee[3] and Seung-Woo Hong[4]

[1] Department of Energy Science, Sungkyunkwan University, Suwon, South Korea
[2] Research Institute of Agriculture and Life Sciences, College of Agriculture and Life Sciences, Seoul National University, Seoul, South Korea
[3] Department of Integrative Biotechnology, Sungkyunkwan University, Suwon, South Korea
[4] Department of Physics, Sungkyunkwan University, Suwon, South Korea
[*] These authors contributed equally to this work.

Corresponding authors
Sukchan Lee, cell4u@skku.edu, sukchan107@gmail.com
Seung-Woo Hong, swhong@skku.ac.kr

## ABSTRACT

Long non-coding RNAs (lncRNAs) play an important role in regulating many biological processes. In this study, tomato seeds were first irradiated by neutrons. Eight tomato mutants were then selected and infected by *Tomato yellow leaf curl virus* (TYLCV). RNA sequencing followed by bioinformatics analyses identified 1,563 tomato lncRNAs. About half of the lncRNAs were derived from intergenic regions, whereas antisense lncRNAs accounted for 35%. There were fewer lncRNAs identified in our study than in other studies identifying tomato lncRNAs. Functional classification of 794 lncRNAs associated with tomato genes showed that many lncRNAs were associated with binding functions required for interactions with other molecules and localized in the cytosol and membrane. In addition, we identified 19 up-regulated and 11 down-regulated tomato lncRNAs by comparing TYLCV infected plants to non-infected plants using previously published data. Based on these results, the lncRNAs identified in this study provide important resources for characterization of tomato lncRNAs in response to TYLCV infection.

## INTRODUCTION

Several of the numerous RNAs that are transcribed by plant genomes, such as messenger RNAs, are translated into proteins. However, others without coding capacity, including non-coding RNAs (ncRNAs), are abundantly present in plant cells (*Fatica & Bozzoni, 2014*). Long non-coding RNAs (lncRNAs) are defined as non-coding RNAs longer than 200 nucleotides (nt) (*Mercer, Dinger & Mattick, 2009*). Due to the rapid advance of next-generation sequencing (NGS) techniques and bioinformatics tools, a large number

of lncRNAs have been identified from various organisms such as animals and plants (*Sun et al., 2017b*). However, lncRNAs have been identified in a limited number of plant species such as *Arabidopsis thaliana*, medicago, potato, rice, strawberry, tomato (*Solanum lycopersicum*), and maize (*Guo & Liu, 2017*; *Kang et al., 2017*; *Nejat & Mantri, 2017*; *Scarano, Rao & Corrado, 2017*; *Zheng et al., 2017*; *Zhu et al., 2017*). Furthermore, studies revealing functional roles of identified lncRNAs in plants are rare. A few studies have shown that lncRNAs might be involved in regulation of several biological functions, such as scaffolding of multiple proteins and gene expression (*Ransohoff, Wei & Khavari, 2017*). In addition, some studies have demonstrated the involvement of plant lncRNAs in biotic and abiotic stresses. For example, previous studies using strand-specific RNA-sequencing has identified several lncRNAs in *Arabidopsis* (*Zhu et al., 2014*) and banana (*Li et al., 2017*), which might be responsible to infection of *Fusarium oxysporum*. Another study has identified DROUGHT INDUCED lncRNA (DRIR) from *Arabidopsis*, which plays a role in responses of drought and salt stress (*Qin et al., 2017*).

Plant transcriptomes can be affected by diverse environmental stimuli such as biotic and abiotic stresses (*Nejat & Mantri, 2017*). Of the known and diverse irradiations, neutron irradiation can have an effect on the genome and transcriptome of living organisms. Furthermore, there are thousands of low energy neutrons in our natural environment, mostly originating from cosmogenic neutron irradiation (*Lal, 1987*). A short duration of cosmogenic neutron irradiation does not significantly damage living organisms; however, accumulated neutron radiation over a long time could be harmful (*Bowlt, 1994*). Recently, neutron radiation has been used to generate deletion mutant populations in diverse plant species, such as barley, rice, pepper, sesame, and *Arabidopsis*, due to its efficient mutagenesis (*Ahloowalia & Maluszynski, 2001*). In addition, those neutron radiation contributes to the evolution of plants on the earth. Changes of plant genomes by neutron irradiation is not always harmful. Sometimes, mutations caused by neutron irradiation in a plant could provide the resistance against a specific biotic stress such as virus infection.

The tomato is an economically important crop as well as a model plant for plant science. *Tomato yellow leaf curl virus* (TYLCV) is one of the serious pathogens causing heavy economical losses. TYLCV in the genus *Begomovirus* is a circular DNA virus and has a broad range of hosts (*Moriones & Navas-Castillo, 2000*; *Polston & Lapidot, 2007*). To date, many studies have been conducted to find tomato cultivars resistance to identify resistance genes to TYLCV (*Ji, Schuster & Scott, 2007*; *Ji et al., 2009*; *Zamir et al., 1994*). Furthermore, a few recent studies have shown that some lncRNAs are involved in TYLCV defense mechanisms in the tomato. For instance, two different lists of lncRNAs associated with TYLCV infection have been identified from resistant and susceptible tomato plants, respectively (*Wang et al., 2018a*; *Wang et al., 2015*).

In this study, mutagenesis was performed on tomato seeds by neutron irradiation. The irradiated tomato plants were infected by TYLCV to select tomato mutants showing resistance against TYLCV infection. To reveal the functional roles of lncRNAs against TYLCV infection in neutron irradiated tomato plants, we conducted RNA-Sequencing (RNA-Seq) for the eight selected tomato mutants. As a result, we identified 1,563 tomato lncRNAs using RNA-Seq and bioinformatics analyses. Furthermore, we characterized the

functional roles of the identified lncRNAs and revealed differentially expressed lncRNAs in response to TYLCV infection using public available data.

## MATERIALS AND METHODS

### Neutron irradiation, plant growth, and TYLCV infection

We used seeds of the tomato cultivar 'Seokwang,' which is susceptible to TYLCV. Two different seed conditions, dry and presoaked, were subjected to neutron irradiation using the MC_50 cyclotron at the Korea Institute of Radiological and Medical Science (KIRAMS) in Seoul, Korea. The presoaked seeds contain a higher portion of oxygen and hydrogen as compared to dry seeds. In particular, it is known that the oxygen can interact with the neutron during irradiation process producing the rapid reactive oxygen species (ROS). ROS plays a pivotal role as a signaling molecule in plants involved in pathogen defense (*Apel & Hirt, 2004*). Thus, it will be interesting to examine possible effects of ROS in the plant transcriptome. The tomato seeds were subjected to neutron irradiation by proton bombardment of beryllium at 40 MeV energy and a 20 µA current. Two different irradiation times of 30 min and 90 min were applied. Neutron-irradiated tomato seeds were grown in a growth chamber under a 16 h light/8 h dark illumination time (*Cheng & Edwards, 1991*). The five-week-old tomato seedlings were infected by TYLCV using a whitefly (*Bemisia tabaci*) vector. TYLCV infected tomato plants were grown for seven weeks in a growth chamber. The general experimental scheme is drawn in Fig. 1A.

### Sample collection and total RNA extraction

Leaf samples from individual tomato mutants showing TYLCV disease symptoms were harvested and immediately frozen in liquid nitrogen. Each sample was examined for TYLCV infection by PCR using TYLCV specific primers. Total RNA was extracted using a Qiagen RNeasy Plant Mini Kit (Qiagen GmbH, Hilden, Germany) following the manufacturer's instructions. DNase I was used to digest genomic DNA in the extracted RNA. The quantity and quality of RNA were measured by an Eppendorf BioPhotometer (MedWOW Ltd., Istanbul, Turkey) and agarose gel electrophoresis.

### Library preparation and RNA sequencing

The poly(A) RNA libraries were prepared using an NEB Next Ultra RNA Library Prep Kit for Illumina (New England BioLabs Inc., Ipswich, MA, USA) according to manufacturer's instructions. The prepared libraries were analyzed by a 2100 Bioanalyzer instrument (Agilent Genomics, Waldbronn, Germany) to measure quality. A total of eight libraries were paired-end (101 bp × 2) sequenced by an Illumina HiSeq 2000 system at Macrogen, Seoul, Korea. The raw sequenced data from this study were deposited in the SRA database at the National Center for Biotechnology Information with the following accession numbers, SRR6019475–SRR6019483. The detailed information of samples is provided in Table 1.

### Assembly of tomato transcripts using RNA-Seq data

All raw data from eight libraries were aligned using Tophat 2.1.1 to the tomato reference genome (ITAG2.4_genomic.fasta) from the International Tomato Genome

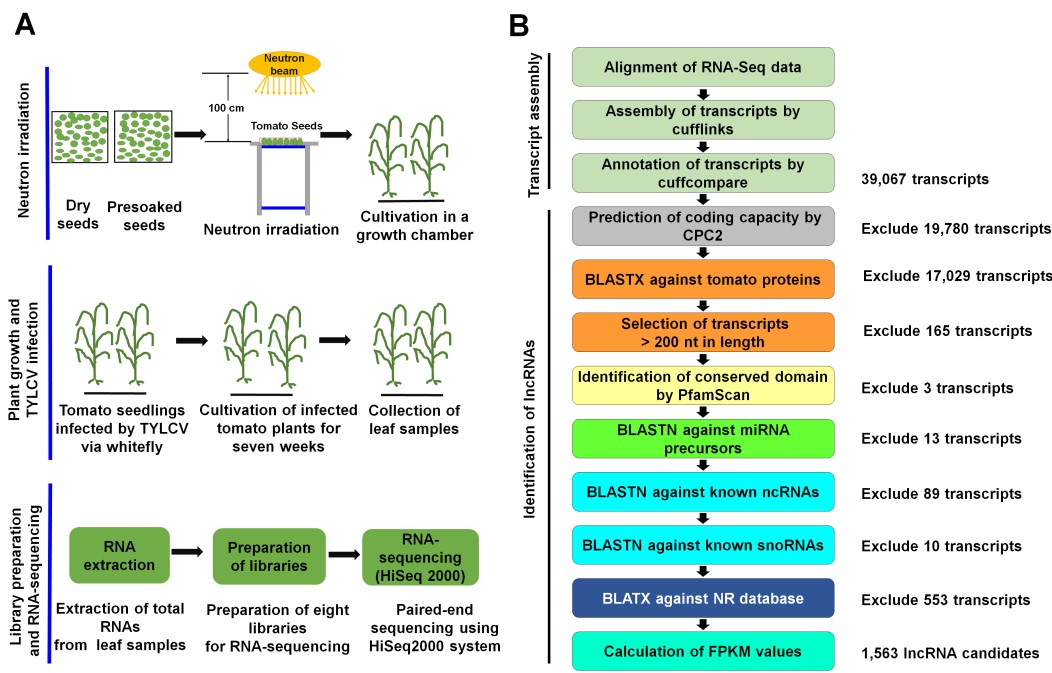

**Figure 1** **Schematic workflow to identify tomato lncRNAs.** (A) Experimental workflow for neutron irradiation, TYLCV infection, and RNA-Seq. The detailed experimental processes have been shown. First, tomato seeds were divided into two groups. Presoaked seeds placed in distilled water for eight hours. Second, after irradiation, the seeds were cultivated in a green chamber and then infected by TYLCV using whitefly. Third, total RNAs were extracted from samples, followed by library preparation and RNA sequencing. (B) Bioinformatic procedures to identify lncRNAs using RNA-Seq data. Paired-end sequence data from eight libraries were subjected to transcript assembly using Tophat and Cufflinks. For each step, the number of excluded transcripts was provided. Full nucleotide sequences and detailed information for 1,563 lncRNAs can be found on the figshare website (https://doi.org/10.6084/m9.figshare.5914396.v1).

**Table 1** **Different samples are denoted by symbols indicating the irradiation time (30 or 90 min), seeds conditions (dry and presoaked), and individual plants (1 and 2).**

| Sample | Seed condition | Neutron irradiation time | TYLCV infection | Protocol 1 | Protocol 2 | Protocol 3 | Data |
|---|---|---|---|---|---|---|---|
| 30D1 | Dry | 30 min | TYLCV infected | Leaf tissues | RNA extraction | Library preparation and RNA-Seq | SRR6019475 |
| 30D2 | Dry | 30 min | TYLCV infected | Leaf tissues | RNA extraction | Library preparation and RNA-Seq | SRR6019476 |
| 30S1 | Presoaked seeds | 30 min | TYLCV infected | Leaf tissues | RNA extraction | Library preparation and RNA-Seq | SRR6019477 |
| 30S2 | Presoaked seeds | 30 min | TYLCV infected | Leaf tissues | RNA extraction | Library preparation and RNA-Seq | SRR6019478 |
| 90D1 | Dry | 90 min | TYLCV infected | Leaf tissues | RNA extraction | Library preparation and RNA-Seq | SRR6019479 |
| 90D2 | Dry | 90 min | TYLCV infected | Leaf tissues | RNA extraction | Library preparation and RNA-Seq | SRR6019480 |
| 90S1 | Presoaked seeds | 90 min | TYLCV infected | Leaf tissues | RNA extraction | Library preparation and RNA-Seq | SRR6019481 |
| 90S2 | Presoaked seeds | 90 min | TYLCV infected | Leaf tissues | RNA extraction | Library preparation and RNA-Seq | SRR6019482 |

Sequencing Project (*Trapnell et al., 2012*). In addition, the Gene Transfer Format (ITAG2.4_gene_models.gff3) file was used for alignment using Tophat resulting in a single BAM file. The BAM file was subjected to Cufflinks 1.3.0 (*Trapnell et al., 2010*) to assemble transcripts. Finally, Cuffcompare was used to annotate lncRNAs (*Trapnell et al., 2010*). We obtained assembled transcripts using Cufflinks. The assembled transcripts were subjected to Cuffcompare to annotate the assembled transcripts by comparing the assembled tomato transcripts to the tomato reference annotation. After that, all transcripts including lncRNAs were annotated. Based on annotation, identified lncRNAs were categorized as intergenic regions, antisense, overlapping, and intronic regions.

## Identification of tomato lncRNAs

As shown in Fig. 1B, several steps were used to predict novel lncRNAs from the assembled transcripts. First, we extracted assembled transcripts in a fasta format using the gffread program (*Weirick et al., 2015*). The assembled transcripts were subjected to the Coding Potential Calculator 2 (CPC2) to predict the coding capacity of each transcript with default parameters (*Kang et al., 2017*). Only transcripts labelled with ''noncoding'' predicted by CPC2 were included for further analyses. Second, the assembled transcripts were subjected to BLASTX (version 2.2.31) against a tomato protein database with an evalue of $1e-3$ as a cutoff to exclude transcripts with a shared sequence similarity to known tomato proteins. In addition, the transcripts with lengths of less than 200 nt were discarded (*Nagata et al., 2004*). Third, the transcripts were subjected to PfamScan (version 31.0) with default parameters against the hmmer protein database (version 3.0b2) to identify those containing conserved protein domains (*Li et al., 2016*). Fourth, results from CPC2, PfamScan, and BLASTX were compared to exclude transcripts with coding capacities. Fifth, we excluded transcripts that showed sequence similarity to known miRNA precursors. For this, BLASTN with an evalue of $1e-$ as a cutoff was conducted against the plant miRNA precursors, house-keeping ncRNAs, small nucleolar RNAs (snoRNAs), and non-redundant (NR) protein databases (*Liu et al., 2005*; *Pruitt, Tatusova & Maglott, 2005*; *Xiao et al., 2009*). Finally, we calculated the fragments per kilobase of transcript per million mapped reads (FPKM) to identify lncRNAs using the Binary version of Sequence Alignment (BWA) followed by the BBMap program with default parameters (*Li & Durbin, 2010*; *Rodríguez-García, Sola-Landa & Barreiro, 2017*). The transcripts in which FPKM values were greater than one were considered putative lncRNAs.

## Functional classification of lncRNAs and gene ontology (GO) enrichment analysis

To obtain a broad functional overview of identified lncRNAs, we predicted targets of identified lncRNAs by BLASTN search against tomato mRNA sequences using an e-value of $1e-$ as a cutoff. Finally, we selected 794 lncRNAs associated with tomato mRNAs. The selected tomato gene sequences were blasted against *Arabidopsis* genes (https://www.arabidopsis.org/) to obtain known functions. A majority of tomato genes were converted to the corresponding *Arabidopsis* locus. Using *Arabidopsis* loci, we conducted GO enrichment analysis using the GOEAST program with default parameters (*Zheng &*

**Table 2  Summary of sequence alignment for eight libraries from the tomato genome.**

| Library name | No. of raw sequence reads | No. of unmapped reads | No. of mapped reads |
|---|---|---|---|
| 30D1 | 15,363,905 | 1,200,998 (7.8%) | 14,162,907 (92.2%) |
| 30D2 | 16,660,349 | 1,446,788 (8.7%) | 15,213,561 (91.3%) |
| 30S1 | 16,851,416 | 2,027,109 (12.0%) | 14,824,307 (88.0%) |
| 30S2 | 16,342,941 | 1,432.998 (8.8%) | 14,909,943 (91.2%) |
| 90D1 | 14,248,247 | 1,079,660 (7.6%) | 13,168,587 (92.4%) |
| 90D2 | 17,578,965 | 1,305,116 (7.4%) | 16,273,849 (92.6%) |
| 90S1 | 15,600,291 | 1,586,768 (10.2%) | 14,013,523 (89.8%) |
| 90S2 | 16,239,706 | 1,264,170 (7.8%) | 14,975,536 (92.2%) |
| Total | 128,885,820 | 11,343,607(8.8%) | 117,542,213 (92.2%) |

*Wang, 2008*). The GO terms obtained for tomato genes possessing lncRNAs were classified according to biological process, molecular function, and cellular component. The REVIGO program was used for visualization of enriched GO terms (*Supek et al., 2011*).

## Expression profiles for 1,563 lncRNAs in response to TYLCV infection

To establish the expression profiles for the 1,563 lncRNAs in this study, we used previously published RNA-Seq data (PRJNA291401) (*Wang et al., 2015*). The published RNA-Seq data consisted of three mock and three TYLCV infected samples. The identified 1,563 lncRNAs were used as reference sequences. Raw sequence reads from the published RNA-Seq data were mapped on the 1,563 lncRNAs using Tophat (*Trapnell et al., 2012*). Transcript assembly was conducted by Cufflinks. TYLCV infected samples were compared to mock samples to identify differentially expressed genes (DEGs) using Cuffdiff based on a $p$-value less than 0.05 and a $\log_2$(fold change) greater than one.

## RESULTS

### Assembly of tomato transcripts using eight RNA-Seq data

To maximize the identification of lncRNAs, we combined all raw data from eight libraries. Raw sequence reads from eight different libraries were subjected to mapping on the reference tomato genome. Of the 128 million reads from eight libraries, almost 117 million (92.2%) were mapped to the tomato genome, whereas about 11 million (8.8%) were not mapped (Table 2). The number of mapped reads ranged from 14,824,307 (30S1) to 16,273,849 (90D2), whereas the portion of unmapped reads ranged from 7.4% (90D2) to 12.0% (30S1). These mapped reads were used for transcript assembly by Cufflinks, resulting in a total of 39,067 transcripts (Fig. 1B).

### Identification of tomato lncRNAs

To identify putative tomato lncRNAs, nine different processes were applied (Fig. 1B). First, 19,780 (50.63%) transcripts with coding capacity were excluded by the CPC2 program, and then a BLASTX search against tomato proteins excluded 17,029 (43.58%) transcripts. In total, 165 transcripts with lengths less than 200 nt were excluded. PfamScan was conducted

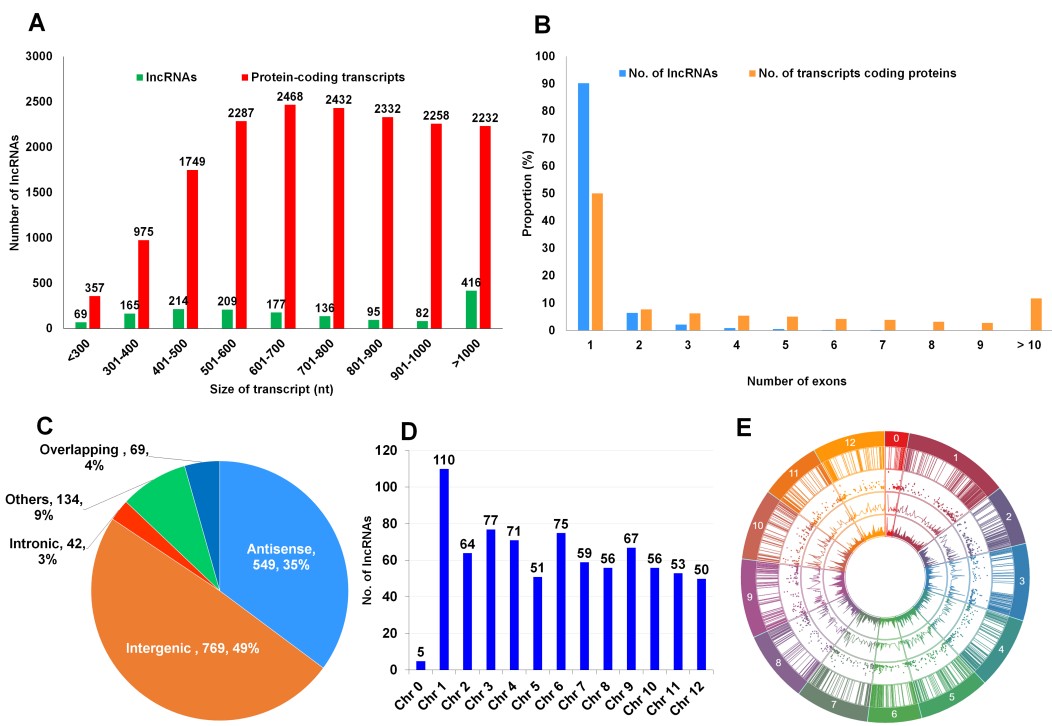

**Figure 2** **Classification of identified tomato lncRNAs.** (A) Size distributions of identified lncRNAs and protein-coding transcripts in this study are visualized by green and red bars, respectively. (B) The proportions of exons associated with lncRNAs and protein-coding transcripts are indicated by blue and orange bars, respectively. (C) Categories of identified lncRNAs. (D) Number of lncRNAs identified on each tomato chromosome. Chromosomal distribution of the identified lncRNAs except those derived from intergenic regions. (E) Chromosomal positions of identified lncRNAs visualized by a circos diagram.

to exclude three transcripts encoding conserved protein domains. This was followed by a BLASTN search against a miRNA precursor database, which excluded 13 transcripts. In addition, 89 and 10 transcripts were excluded by a BLASTN search for showing sequence similarity to known ncRNAs and snoRNAs, respectively. Finally, the remaining transcripts were subjected to a BLASTX search against the NR protein database in NCBI, which excluded 553 transcripts. As a result, we obtained 1,563 lncRNAs from eight tomato samples (Table S1).

## Classification of identified tomato lncRNAs

The lengths of the identified lncRNAs ranged from 201 nt to 4,647 nt. About 26% (416 lncRNAs) of identified lncRNAs were more than 1,000 nt, while 74% of identified lncRNAs were less than 1,000 nt. However, most protein coding transcripts (94.3%) were less than 1,000 nt in size. Among the lncRNAs less than 1,000 nt, lengths ranged from 301 to 800 nt (Fig. 2A). We compared the length distribution of identified lncRNAs between our study and a previous study (*Wang et al., 2015*). Both studies showed similar length distribution of identified lncRNAs. In particular, the length of lncRNAs ranged from 400 nt to 500 nt was the highest proportion in both studies. In addition, we examined the

number of exons in lncRNAs as well as protein coding transcripts (Fig. 2B). Most lncRNAs (90.2%) were derived from a single exon, and there were two lncRNAs, TCONS_00021506 and TCONS_00021507, derived from six and seven exons, respectively. However, the functions of corresponding genes were unknown. In comparison, half of the protein coding transcripts contain a single exon, whereas 11.7% of the protein coding transcripts have more than 10 exons.

Next, we categorized the identified lncRNAs. Most lncRNAs (49%) were derived from intergenic regions, followed by antisense lncRNAs (35%), overlapping (4%), and intronic regions (3%) (Fig. 2C). Apart from the 769 lncRNAs derived from intergenic regions, 794 lncRNAs were associated with a gene. We further classified the 794 lncRNAs according to chromosome. Five lncRNAs were not assigned to any chromosome. It seems that they were located on random scaffold. With the exception of 110 lncRNAs on chromosome 1, the number of lncRNAs on each chromosome ranged from 50 (Chromosome 12) to 77 (Chromosome 3), as shown in Fig. 2D. The positions of identified lncRNAs on each tomato chromosome are indicated in a circos diagram (Fig. 2E). In order to visualize the distribution of identified lncRNAs on different chromosomes, a combination of different graphs including bar plots and volcano plots were used. Interestingly, most lncRNAs were highly enriched at the beginning and the ending of each chromosome. There were relatively few lncRNAs located in the middle of individual chromosome.

## Expression profiles of identified lncRNAs in eight samples

We examined the expression of 1,563 lncRNAs in eight different samples by calculating the FPKM values. Some lncRNAs were not expressed in eight conditions; however, most lncRNAs were highly expressed. In order to examine the distribution of the expression values for all identified lncRNAs, we generated a box plot using $\log_{10}$ converted FPKM values (Fig. 3A). Although standard deviation showed a high degree of difference in each sample, minimum, median, and maximum values among samples did not show significant difference. Next, we conducted PCA analysis to cluster eight conditions based on gene expression of lncRNAs. Our result showed that seven conditions except the condition for soaked sample for 30 min (30S1) were clustered together (Fig. 3B).

We next examined the expression profile of 1,563 lncRNAs in response to TYLCV infection using previously published data (*Wang et al., 2015*). As a result, 915 lncRNAs out of 1,563 were expressed in mock and TYLCV infected sample (Table S2). A volcano plot showed that the number of up-regulated lncRNAs (19 lncRNAs) was higher than that of down-regulated lncRNAs (11 lncRNAs) (Fig. 3C). Expression of 30 differentially expressed lncRNAs was visualized by a heat map (Fig. 3D). Unfortunately, the functions of those 30 differentially expressed lncRNAs are not currently known due to lack of functional studies associated with tomato lncRNAs.

## Functional classification of tomato genes associated with identified lncRNAs

Of the 1,563 lncRNAs identified, 794 were associated with tomato genes (Table S3). In order to reveal the functional roles of tomato genes associated with lncRNAs, we examined

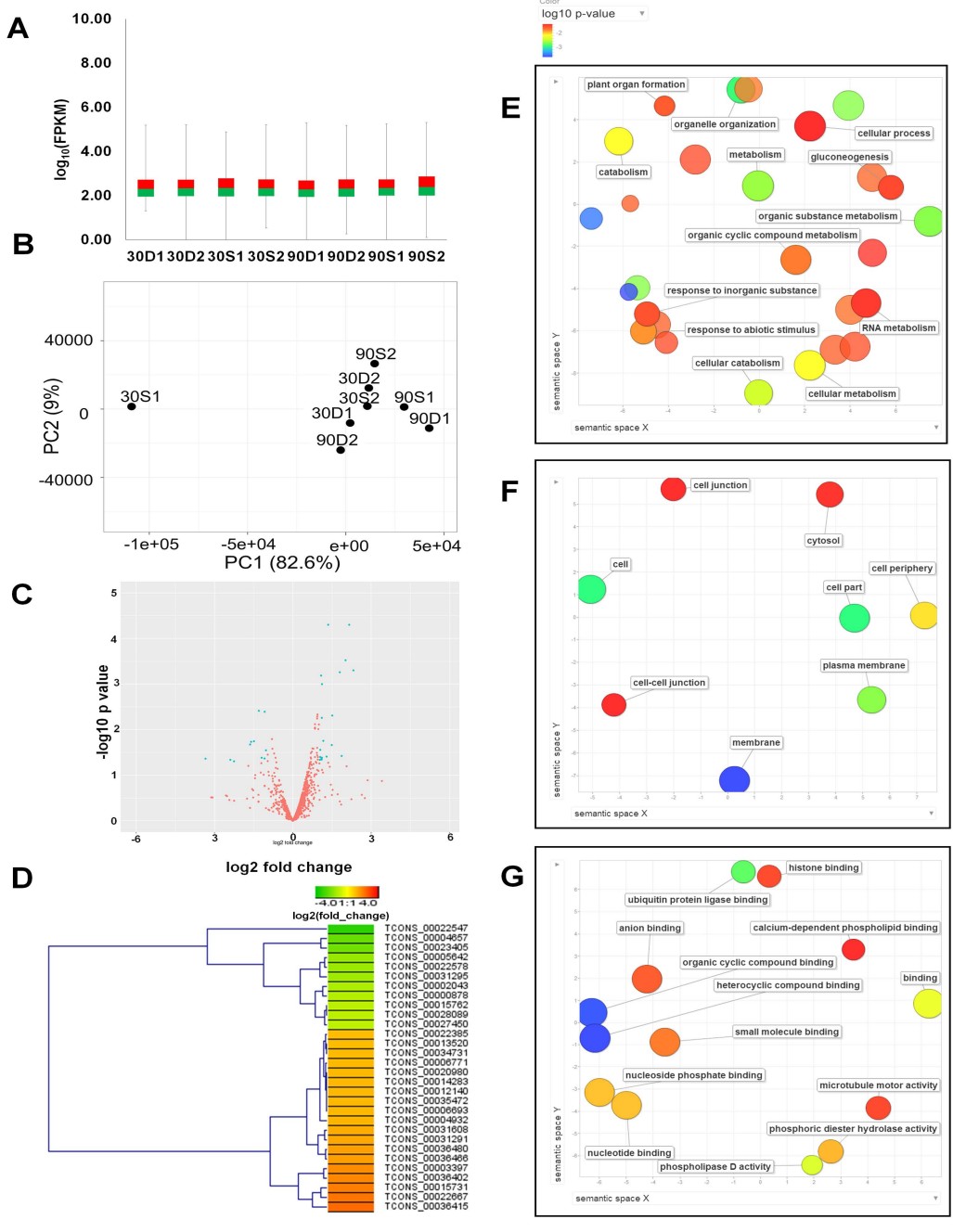

**Figure 3 Expression profiles of identified lncRNAs and their functional classification.** (A) Box plot showing the overall distribution of lncRNAs expression (log10 converted FPKM values) in eight samples. (B) PCA analysis of eight conditions based on the expression of lncRNAs. FPKM values of lncRNAs in eight conditions were subjected to PCA analysis using ClustVis program (https://biit.cs.ut.ee/clustvis/). (C) Volcano plot illustrating the distribution of *p*-values and fold changes for expression of lncRNAs in response to TYLCV infection compared to mock samples. Blue colored dots indicate differentially expressed genes (DEGs). (D) The expression levels of differentially expressed lncRNAs in response to lncRNAs visualized by a heat map. Enriched GO terms of the identified lncRNAs based on biological process (E), cellular component (F), and molecular function (G).

GO using corresponding *Arabidopsis* genes. GO enrichment analyses revealed that 47 GO terms (biological process), eight GO terms (cellular component), and 14 GO terms (molecular function) were highly enriched in tomato gene associated lncRNAs (Table S4). In relation to biological processes, GO terms associated with positive regulation of the abscisic acid-activated signaling pathway, organic substance metabolism, gluconeogenesis, organelle organization, and cellular catabolism were highly enriched (Fig. 3E). Based on cellular component analysis, GO terms related to cytosol, membrane, and cell junction were highly enriched (Fig. 3F). Interestingly, according to molecular function, many GO terms were associated with binding. For example, heterocyclic compound binding, phospholipase D activity, and ubiquitin protein ligase binding were frequently identified (Fig. 3G).

### Identification of target tomato mRNAs of the 1,563 lncRNAs

GO enrichment analysis revealed that many genes associated with lncRNAs have binding functions important for interaction of lncRNAs with other molecules such as RNAs and proteins. We identified target tomato mRNAs that showed strong sequence similarity to the lncRNAs identified by BLASTN search (Table S5).

Of the lncRNAs identified, 824 showed sequence similarity to at least one tomato mRNA. For instance, 566 lncRNAs has a single target, whereas two lncRNAs showed sequence similarity to 10 different nucleotide sequences (Fig. 4A). The lncRNA TCONS_00000794 (Fig. 4B), which was 2,297 nt in length, displayed 10 different regions of six genes. The lncRNA TCONS_00003273, which was 2,081 nt in length, displayed 10 different regions of six genes (Fig. 4C).

Of 30 differentially expressed lncRNAs, six lncRNAs showed sequence similarity to tomato mRNAs; however, functions of only three corresponding mRNAs are known. For instance, the lncRNA TCONS_00005642, which was down-regulated by TYLCV infection, is associated with BHLH transcription factor (Solyc02g063440.2). Two lncRNAs, TCONS_00020980 and TCONS_00035472, showed sequence similarity to a gene coding pyruvate decarboxylase 2 (Solyc06g082140.2) and a gene coding ariadne-like ubiquitin ligase (Solyc11g008590.1), respectively. Both lncRNAs were up-regulated by TYLCV infection.

## DISCUSSION

In this study, we identified tomato lncRNAs that might be associated with at least two different factors. One is neutron irradiation, which causes mutagenesis in the tomato genome, and the other is TYLCV infection, which could change the tomato transcriptome. The irradiated tomato plants showed no significant disease symptoms after TYLCV infection. This result indicates that the genomes of tomato plants could be mutated by neutron irradiation. Thus, the transcriptional regulation in those tomato plants could be changed. We examined lncRNAs to check the change of transcriptome caused by neutron irradiation. Moreover, there were several previous studies identifying lncRNAs in response to TYLCV infection, which facilitates comparison of the lncRNAs among different studies. Since the release of the draft tomato genome sequence (*The Tomato Genome Consortium, 2012*), several studies have identified diverse tomato lncRNAs. To date, the identified

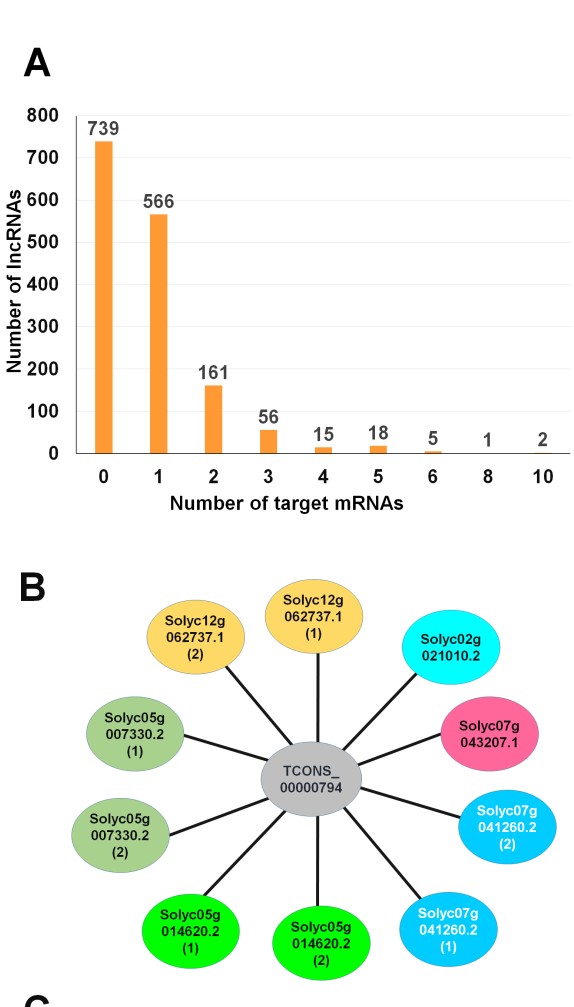

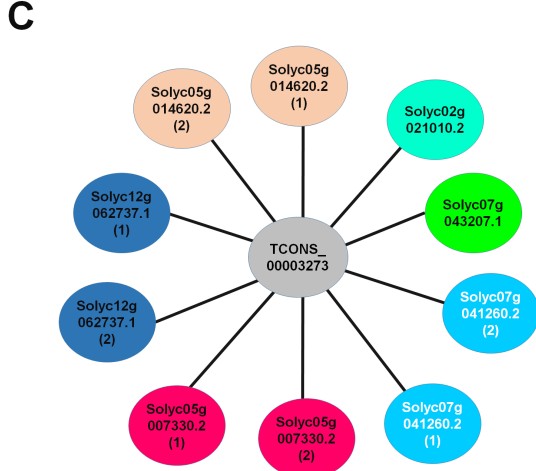

**Figure 4    Identification of target mRNAs for the identified lncRNAs by sequence similarity.** (A) Number of target mRNAs for individual lncRNA revealed by a BLASTN search. The possible interactions of mRNAs with two selected lncRNAs, TCONS_00000794 (B) and TCONS_00003273 (C). Some lncRNAs showed sequence similarity to two regions of a mRNA indicated by 1 and 2.

tomato lncRNAs are related with tomato fruit development (*Scarano, Rao & Corrado, 2017*; *Sun & Xiao, 2015*; *Wang et al., 2018b*; *Zhu et al., 2015*) and TYLCV infection (*Wang et al., 2018a*; *Wang et al., 2015*). It seems that fruit development and TYLCV infection, which was included in our study, are two of the most important biological processes and pathogen responses associated with lncRNAs in tomato plants.

Compared to two other studies associated with lncRNAs in response to TYLCV infection (*Wang et al., 2018a*; *Wang et al., 2015*), the number of lncRNAs identified in our study (1,563 lncRNAs) is comparable (1,565 and 2,056 lncRNAs, respectively). A recent study demonstrated that most lncRNAs with low expression are tissue-specific, whereas constitutively expressed lncRNAs are highly conserved in plant species (*Deng et al., 2018*). Thus, it is important to identify lncRNAs from diverse plant samples because some could be highly regulated by specific environmental factors such as stress conditions, tissues, and developmental stage. For instance, a recent study combining 134 RNA-Seq data has identified 70,635 lncRNAs (*Wang et al., 2018b*), while the number of lncRNAs identified in other studies ranged from 1,565 (*Wang et al., 2015*) to 10,774 (*Scarano, Rao & Corrado, 2017*).

Although we used a tomato cultivar susceptible to TYLCV, most neutron-irradiated plants showed reduced disease symptoms or no symptoms. Thus, the expression of lncRNAs in our study might be associated with a previous study using a tomato cultivar (CLN2777A) resistant to TYLCV (*Wang et al., 2015*). The expression profile of the 1,563 lncRNAs in our study using the previous study showed that about 59% of lncRNAs (915 lncRNAs) were transcribed. We hypothesize cautiously that different genetic backgrounds caused by neutron irradiation change the transcription of several lncRNAs, although both studies performed RNA-Seq followed by TYLCV infection.

Although RNA-Seq facilitates the identification of numerous lncRNAs in many plant species, only a few lncRNAs were annotated and characterized (*Liu, Wang & Chua, 2015*). For instance, several studies showed that some lncRNAs are involved in biotic and abiotic stresses (*Nejat & Mantri, 2017*). An *Arabidopsis* lncRNA known as ELENA1 interacts with MED19a to regulate PR1 expression functions in increased resistance against *Pseudomonas syringae* pv tomato DC3000 (*Seo et al., 2017*). As shown in previous studies, it is very important for lncRNAs to interact with DNAs to form a stable RNA-DNA complex to control the transcriptional activities of target genes (*Liu, Wang & Chua, 2015*). Previous studies also demonstrated that functions of lncRNAs rely on their binding properties with other nucleic acids and proteins (*Marchese, Raimondi & Huarte, 2017*; *Sun, Ali & Moran, 2017a*). For example, many lncRNAs contain several functional regions which are required for interaction with other factors such as ribonucleoproteins and diverse RNA-binding proteins. Similarly, we found that many lncRNA associated mRNAs have binding functions such as nucleotide, small molecule, anion, and histone binding. This result directly suggests that the interaction of lncRNAs with target molecules is an important step in their transcriptional regulation of diverse biological processes, as shown recently (*Shi et al., 2017*).

Cytoplasm is the place where many lncRNAs are activated (*Rashid, Shah & Shan, 2016*). Similarly, GO enrichment analysis showed that the lncRNAs in our study were associated

with the cytosol and plasma membrane, suggesting that these two cellular components are important places for lncRNAs. In addition, we found that many mRNA targets of lncRNAs were targeted to organelles such as plastids and mitochondria suggesting strong involvements of lncRNAs in organelle biogenesis. A recent study suggests that involvement of lncRNAs not only in nucleus but also in outside of the nucleus (*Krause, 2018*). Of identified biological processes related to target mRNAs of lncRNAs, functions associated with hormone metabolism, lignin metabolism, developmental processes such as post-embryonic development and developmental process involved in reproduction were highly enriched. Similarly, recent studies also demonstrated the involvement of lncRNAs in metabolisms (*An et al., 2018*; *Lu et al., 2016*).

Most known lncRNAs were derived from intergenic regions. For example, a previous study showed that 89% of tomato lncRNAs were derived from intergenic regions, while antisense lncRNAs accounted for only 10% (*Wang et al., 2018b*). In contrast, our study revealed that 49% of lncRNAs was derived from intergenic regions, and antisense lncRNAs accounted for 35%. Moreover, the sizes of the lncRNAs in our study were relatively smaller than that of a previous study (*Wang et al., 2015*). We hypothesize cautiously that mutagenesis caused by neutron irradiation interferes with the transcriptional regulation of lncRNA expression, resulting in fewer small-sized lncRNAs.

Based on sequence similarity BLASTN search, we found several mRNA targets corresponding to identified lncRNAs. Most lncRNAs have a single mRNA target but some lncRNAs have multiple mRNA targets which might be members in the same gene family with strong sequence similarity. Therefore, it is likely that the lncRNAs with multiple target regulate expression of target mRNAs simultaneously.

## CONCLUSION

The present study provides a comprehensive bioinformatics analysis of lncRNAs in tomato plants irradiated by neutrons, followed by TYLCV infection. Mutagenesis caused by neutrons influences the transcription of many lncRNAs with shorter lengths and increases the number of antisense lncRNAs. Furthermore, we identified key lncRNAs that are important for TYLCV infection. Based on these results, the lncRNAs identified in this study provide important resources for characterization of tomato lncRNAs in response to diverse stimuli.

## ACKNOWLEDGEMENTS

The authors express their sincere thanks to the staff of the MC-50 Cyclotron Laboratory (KIRAMS) for the excellent operation and their support during the experiment.

### Funding

This work was supported by the National Research Foundation of Korea grant funded by the Korean government (MSIT) (NRF-2017R1A2B2005117 to S.L.), (NRF-2018M7A1A1072274 to S.-W.H.), and (NRF-2016R1D1A1B03935429 to V.C.). The funders had no role in study design, data collection and analysis, decision to publish, or preparation of the manuscript.

### Grant Disclosures

The following grant information was disclosed by the authors:
National Research Foundation of Korea (NRF).
Korean goverment (MSIT): NRF-2017R1A2B2005117, NRF-2018M7A1A1072274, NRF-2016R1D1A1B03935429.

### Competing Interests

The authors declare there are no competing interests.

### Author Contributions

- Yujie Zhou conceived and designed the experiments, performed the experiments, analyzed the data, prepared figures and/or tables, authored or reviewed drafts of the paper, approved the final draft.
- Won Kyong Cho conceived and designed the experiments, analyzed the data, contributed reagents/materials/analysis tools, prepared figures and/or tables, authored or reviewed drafts of the paper, approved the final draft.
- Hee-Seong Byun performed the experiments, approved the final draft.
- Vivek Chavan analyzed the data, approved the final draft.
- Eui-Joon Kil performed the experiments, contributed reagents/materials/analysis tools, approved the final draft.
- Sukchan Lee and Seung-Woo Hong conceived and designed the experiments, contributed reagents/materials/analysis tools, authored or reviewed drafts of the paper, approved the final draft.

### Data Availability

   The data is available at the NCBI SRA database at the project number PRJNA403767.

### Supplemental Information

Supplemental information for this article can be found online at http://dx.doi.org/10.7717/peerj.6286#supplemental-information.

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
