# Peer review of "Genome-wide identification of long non-coding RNAs in tomato plants irradiated by neutrons followed by infection with Tomato yellow leaf curl virus"

_PeerJ, doi:10.7717/peerj.6286_

## Round 0.1 · original submission · Major Revisions

Your manuscript has been reviewed by two experts, each of whom identify major items that need to be addressed. I agree with their suggestions, please revise the manuscript to address them.

·

Basic reporting

The writing of this paper needs to be improved. Background information is lacking in the the introduction. Additional analysis or more discussion on the differentially expressed lncRNAs should be included.

The abstract is unclear on the rational of this work. It should be more structured into a) why studying long non-coding RNA in tomato irradiated by neutrons followed by yellow leaf curl virus infection; b) the experimental design and analyses; c) the major results and conclusion.

L 31: Are the length of the lncRNAs small or the number of lncRNAs small? Are they only smaller than what was found in one previous study? Was that a tomato study? Please indicate.

L 35: up- and down- regulated lncRNAs in TYLCV infected plants compared to what?

The introduction lacks important information on the background, such as why infect tomato with TYLCV after neutron irradiation, why this needs to be studied? How can lncRNA tell us some information that other study could not, such as mRNA? Are there any other published lncRNA study in tomato? They should be reviewed if there are. I suggest expanding introduction 3rd paragraph with such information. I suggest the last paragraph of introduction to describe the general experiment setup and aim(s) of this study.

L44: there are other types of non-coding RNAs, such as microRNA, please change “known as” to “including”.

L53: the information is limited in terms of what? Please indicate.

L53-L55: Are lncRNA involved in biotic and abiotic stresses? If yes, please include them in the introduction.

Experimental design

The study is interesting but the research question is not very well defined, the reason for why to combine neutron irradiation and TYLCV infection in one study should be included. Methods lacks some details and information to replicate.

Material and Methods are missing details on how different programs were used and parameters for different programs. I am also wondering why you didn’t do de-novo assembly using Trinity, many studies have shown that combining de-novo and reference-based assembly can help to identify more transcripts. Also, did you remove repeats with many more mapping positions on the reference genome? How did you do expression analysis is unclear.

L77: Why to include both dry and presoaked seeds? Please indicate.

L81-L82: Is green chamber also growth chamber? Please be consistent.

L110-L111: Did you use cuffmerge after cufflinks to merge separate assembly into one? As I understand, you generate one set of lncRNA at the end of the analysis instead of eight, where did you conduct merging? It is not reflected in figure 1. Cuffcompare can helps you to compare your assembly to known transcripts and annotate your assembly with difference code. Did you use any of those code to filter your assembly?

L123: I thought you filtered out transcripts from each step individually, as what you showed in figure1, this sentence seems contradictory to your pipeline in figure1. Also for CPC2, is there a threshold value you used? Please indicate if yes.

L125-L127: Please indicate the e-value you used for your BLASTN search.

L129-L131: BWA and BBMap are both alignment tools, I don’t understand why you used two alignment tools to get FPKM values. What’s the distribution of FPKM values look like? How many transcripts did you filter out using FPKM of 1? Could this threshold be too high? Because lncRNAs are known to have low expression.

L134: Did you predict lncRNA target? How? Please describe.

L133: Tomato genome has been published for a long time, there should be published GO term for tomato genes as well. I think you’d better use the GO term for tomato genes for GO enrichment analysis instead of Arabidopsis homolog GO terms.

L146-L147: I am unclear how you did the alignment and quantification of lncRNAs, align lncRNA to lncRNA? What tool did you use for expression analysis? I would suggest using edgeR or Deseq for expression analysis.

Validity of the findings

For the finding, additional analysis or discussion should be included for the differentially expressed lncRNAs in response to TYLCV.

Results are too abstract, lacks detailed description of the findings. I also want to know for the 19 and 11 up or down regulated lncRNAs, what genes are they derived from if they are associated with genes and what target do they have? Since they are TYLCV responsive lncRNAs, they should be examined and discussed in detail.

L173-L196: Did you compare to other studies in respect of the lncRNA length distribution? Please include in the discussion section. “protein coding transcripts”, not “transcripts coding protein”.

L178: for the one that were derived from six and seven exons, I think you need to give some explanation in the discussion section.

L179: Instead of using “Otherwise”, I suggest change to “in comparison”.

L185: Why were five lncRNA not assigned to any chromosome? They were mapped to the reference genome but could not be located on any chromosome? Are they located on random scaffold? Please explain.

L188: There are multiple circles on the circus plot, please describe in detail. Also please describe the distribution pattern in the result section. Are there more lncRNA in certain genomic region?

L191-L196: What are those eight lncRNAs? Which samples did not express them? Did neutron irradiation with different duration of time or seed condition have effect on their expression? Did you do PCA or mds analysis to see the samples clustering pattern based on their expression level of lncRNAs? I am expecting to see same experiment conditions cluster together. Same for the RNA-seq data from other studies, I think you should do PCA analysis to check whether TYLCV treated and untreated samples cluster together. Please include the results in the supplementary material.

L201: Please rewrite this sentence, it is confusing. I think you mean the number of up-regulated lncRNAs was higher, with 19 upregulated whereas 10 down-regulated.

L203-L215: What do the different colors mean in Figure3 D-F?

L209: Could the enrichment of “ABA-activated signaling pathway” be caused by your dry or pre-soaked treatment?

L218-L219: Is this consistent with what was already known? Please describe in the discussion section.

L222-L227: What are the lncRNA mRNA targets? What biological function do they have? Why do some lncRNAs have multiple targets? What are those lncRNAs? What genes are they derived from? All of these should be discussed.

Discussion sections lacks enough discussion about the study findings.

L230-L232: you are dealing with two factors at the same time, why is it important to combine these two factors in one experiment and check lncRNA? Please discuss.

L233-L234: You are missing reference here.

L236-L238: Please give more details on how lncRNA is involved in fruit development and TYLCV infection.

L249-L252: How can you show the “correlation” between your lncRNA with the CLN2777A study? Could it be that neutron-irradiation caused mutation on the susceptible allele, so that TYLCV susceptibility was alleviated?

L252-L256: Was the same genotype used for both RNA-seq experiment? It is possible that different genotypes have different expression of lncRNAs. Is the difference too big compared to other studies? Since only 59% lncRNAs are expressed. Please discuss.

Additional comments

This study is interesting and the data analysis was done in a reasonable way, however, the writing should be improved, and a lot of necessary information is missing, such as background information, detailed description of the result, and enough discussion of the findings. I would also suggest have someone who is more professional in English writing proofread the manuscript.

Reviewer 2 ·

Basic reporting

a. The article is well written with clear and professional English. However there are minor edits need to be done.
for example: line 43 should be re-worded so that it's easier to follow. Line 128 delete "to identify lncRNAs"
b. Other studies of lncRNA on TYLCV in tomato were not mentioned in the introduction. Therefore the knowledge gap filled by this study is not well defined.

Experimental design

a. The hypothesis of this study is not clear. A RNA-seq analysis on the susceptible cultivar infected by TYLCV would enhance the biological interest of this study.
b. The expression analysis of data from Wang, 2015 seemed to be a mere repeat of the original publication. It would be helpful to compare the genotypes used in your study and wang, 2015, and how your differential expression analysis added to the previous publication.

Validity of the findings

In the conclusion, The influence of mutagenesis on the transcription of lncRNAs was not supported by data.

Additional comments

a. Figure 2C needs to define category Overlapping. Figure 2E needs to describe the four tracks.
b. Figure 3D-F need more description.

---

## Round 0.2 · Minor Revisions

Please address the remaining comments from reviewer 1. Also, while you indicate that no GO terms are available for tomato that is incorrect. You can get them from ftp://ftp.solgenomics.net/tomato_genome/annotation/ITAG3.2_release/ITAG3.2_protein_go.tsv

·

Basic reporting

The authors have revised the manuscript, I am OK with most of the responses. However, there are several places that I feel need more revision.

Experimental design

1) cuffcompare is not designed to do expression analysis, please check.

2) Comment 9 (L77): Why to include both dry and presoaked seeds? Please indicate.
Response: We have addressed the reason why we included both dry and presoaked seeds in our study as follows.
In the manuscript: Presoaked seeds contain higher portions of oxygen and hydrogen than dry seeds. In particular, it is known that oxygen can interact with neutrons during the irradiation process to produce reactive oxygen species (ROS). ROS play a pivotal role as signaling molecules in plants involved in pathogen defense (Apel and Hirt 2004). Thus, it will be interesting to examine the possible effects of ROS in the plant transcriptome.
Comment: OK, but I think since you didn’t do expression analysis for your specific samples, the real contribution of this design is to help you get lncRNA from different conditions, because you combined all samples for lncRNA identification.

3) Comment 11 (L110-111): Did you use cuffmerge after cufflinks to merge separate assembly into one? As I understand, you generate one set of lncRNA at the end of the analysis instead of eight, where did you conduct merging? It is not reflected in figure 1. Cuffcompare can helps you to compare your assembly to known transcripts and annotate your assembly with difference code. Did you use any of those code to filter your assembly?
Response: We mapped all raw sequence reads from eight samples on the reference genome using Tophat. After that, a single bam file was processed with Cufflinks. Finally, we obtained 39,067 transcripts by Cuffcompare. We did not use any code to filter our assembly. We have revised the relevant paragraph as follows.
In the manuscript: All raw data from the eight libraries were aligned using Tophat 2.1.1 to the tomato reference genome (ITAG2.4_genomic.fasta) from the International Tomato Genome Sequencing Project (Trapnell et al. 2012). In addition, the Gene Transfer Format (ITAG2.4_gene_models.gff3) file was used for alignment using Tophat, resulting in a single BAM file. Cufflinks 1.3.0 (Trapnell et al. 2010) was used to assemble the BAM file into transcripts. Finally, Cuffcompare was used to annotate lncRNAs (Trapnell et al. 2010).
Comment: OK. Then how did you use cuffcompare to annotate lncRNAs? What kind of insight did you gain from cuffcompare? Please include the details.

4) Comment 16 (L133): Tomato genome has been published for a long time, there should be published GO term for tomato genes as well. I think you’d better use the GO term for tomato genes for GO enrichment analysis instead of Arabidopsis homolog GO terms.
Response: Yes, it is desirable to use tomato genes for GO enrichment instead of Arabidopsis genes. However, there is not a program or database for tomato GO terms, and tomato gene annotation is less comprehensive than that of Arabidopsis. Thus, we conducted GO enrichment analysis using homologous Arabidopsis GO terms.
Comment: I am OK with this. But you could use R package “goseq” for GO enrichment analysis in any organism with published GO annotations, please check this site: http://jnmaloof.github.io/BIS180L_web/2018/05/24/RNAseq-Annotation/.

Validity of the findings

5) Comment: please also include a detailed figure legend to part E of figure A. E.g. what does each track mean?

6) Comment 24 (L191-L196): What are those eight lncRNAs? Which samples did not express them? Did neutron irradiation with different duration of time or seed condition have effect on their expression? Did you do PCA or mds analysis to see the samples clustering pattern based on their expression level of lncRNAs? I am expecting to see same experiment conditions cluster together. Same for the RNA-seq data from other studies, I think you should do PCA analysis to check whether TYLCV treated and untreated samples cluster together. Please include the results in the supplementary material.
Response: As the reviewer suggested, we have conducted PCA analysis to cluster conditions based on gene expression of lncRNAs. Our result showed that seven samples, except the sample soaked for 30 min (30S1), were clustered together. In the RNA-Seq data from other studies, only two conditions (mock and TYLCV infected samples) were used. Instead of carrying out PCA analysis, we generated a heat map showing differentially expressed genes between mock and TYLCV infected samples (Figure 3D).
In the manuscript: Some lncRNAs were not expressed in the eight conditions.
Next, we conducted PCA analysis to cluster the eight conditions based on gene expression of lncRNAs. Our results showed that, except for the conditions where the sample was soaked for 30 min (30S1), seven conditions were clustered together (Figure 3B).
Comment: for the TYLCV infected and mock samples, I thought you had different samples for the same condition. No need to include the “heatmap” if there are only two samples. Also the figure 3 legend is incorrect.

7) Comment 30 (L230-L232): you are dealing with two factors at the same time, why is it important to combine these two factors in one experiment and check lncRNA? Please discuss.
Response: We have included a paragraph in the discussion as follows.
In the manuscript: The irradiated tomato plants showed no significant disease symptoms after TYLCV infection. This result indicates that the genomes of tomato plants were mutated by neutron irradiation. Thus, the transcriptional regulation in these tomato plants might be changed. We examined lncRNAs to confirm the change of transcriptome caused by neutron irradiation. Moreover, several previous studies identified lncRNAs in response to TYLCV infection, which facilitates comparison of the lncRNAs among different studies.
Comment: How did you “confirm the change of transcriptome caused by neutron irradiation”? Maybe because only ~50% lncRNAs identified from your study were expressed in the other study? Please discuss. It would be good if you had TYLCV uninfected sample included in your experiment design, so you would be able to compare the effect of neutron irradiation on TYLCV resistance. Please discuss.

8) Comment 33 (L249-L252): How can you show the “correlation” between your lncRNA with the CLN2777A study? Could it be that neutron-irradiation caused mutation on the susceptible allele, so that TYLCV susceptibility was alleviated?
Response: Yes, you are right. In our study, we used the tomato cultivar Seokwang, which was susceptible to TYLCV after neutron irradiation. The eight selected tomato mutants were resistant to TYLCV or showed weak disease symptoms in response to TYLCV infection. Of course, there were some deviations of TYLCV resistance among different mutants; however, all selected mutants showed alleviated TYLCV susceptibility compared to wild type tomato.
Comment: I still don’t understand how you found correlation. Correlation is a statistical term, you need data to support.

Reviewer 2 ·

Basic reporting

Clarity of the writing has been greatly improved.

Experimental design

No comment.

Validity of the findings

No comment.

---

## Round 0.3 · accepted · Accept

Thank you for submitting your work to PeerJ and to addressing the reviewers comments.

#